# OOSP: Opportunistic Optimization Scheme for Pod Deployment Enhanced with Multilayered Sensing

**DOI:** 10.3390/s24196244

**Published:** 2024-09-26

**Authors:** Joo-Young Roh, Sang-Hoon Choi, Ki-Woong Park

**Affiliations:** 1SysCore Lab, Sejong University, Seoul 05006, Republic of Korea; 2Department of Computer and Information Security, Sejong University, Seoul 05006, Republic of Korea

**Keywords:** microservice architecture, container orchestration, kubernetes, inspection, system optimization

## Abstract

In modern cloud environments, container orchestration tools are essential for effectively managing diverse workloads and services, and Kubernetes has become the de facto standard tool for automating the deployment, scaling, and operation of containerized applications. While Kubernetes plays an important role in optimizing and managing the deployment of diverse services and applications, its default scheduling approach, which is not optimized for all types of workloads, can often result in poor performance and wasted resources. This is particularly true in environments with complex interactions between services, such as microservice architectures. The traditional Kubernetes scheduler makes scheduling decisions based on CPU and memory usage, but the limitation of this arrangement is that it does not fully account for the performance and resource efficiency of the application. As a result, the communication latency between services increases, and the overall system performance suffers. Therefore, a more sophisticated and adaptive scheduling method is required. In this work, we propose an adaptive pod placement optimization technique using multi-tier inspection to address these issues. The proposed technique collects and analyzes multi-tier data to improve application performance and resource efficiency, which are overlooked by the default Kubernetes scheduler. It derives optimal placements based on the coupling and dependencies between pods, resulting in more efficient resource usage and better performance. To validate the performance of the proposed method, we configured a Kubernetes cluster in a virtualized environment and conducted experiments using a benchmark application with a microservice architecture. The experimental results show that the proposed method outperforms the existing Kubernetes scheduler, reducing the average response time by up to 11.5% and increasing the number of requests processed per second by up to 10.04%. This indicates that the proposed method minimizes the inter-pod communication delay and improves the system-wide resource utilization. This research aims to optimize application performance and increase resource efficiency in cloud-native environments, and the proposed technique can be applied to different cloud environments and workloads in the future to provide more generalized optimizations. This is expected to contribute to increasing the operational efficiency of cloud infrastructure and improving the quality of service.

## 1. Introduction

Cloud systems continue to grow, with the global cloud computing market projected to reach USD 678.8 billion by 2024 [1]. This growth is a reflection of the widespread adoption of cloud systems across various industries, as well as the implementation of container technology, which is a key component in building cloud systems. Containers leverage the isolation features of modern operating systems, minimizing resource usage and enhancing deployment flexibility. Microservice architecture, which encapsulates various services within an application in separate containers and deploys them across physical or virtual machine clusters, has become more popular due to its efficiency and flexibility. Many cloud service providers now offer containers-as-a-service (CaaS), simplifying the deployment of containerized applications in the cloud [2]. The growing number of enterprises seeking to avoid vendor lock-in and capitalize on container portability between environments has led to a surge in CaaS adoption. CaaS platforms typically handle authentication, logging, security, monitoring, networking, load balancing, auto-scaling, and continuous integration/continuous delivery (CI/CD). The Cloud Native Computing Foundation (CNCF) [3] defines cloud-native as a new computing paradigm that builds applications using microservice architecture, packages them in containers, and dynamically manages them with orchestrators. Google has operated large-scale container systems for commercial use for a long time. Despite the benefits of cloud computing, the lack of a suitable orchestrator led Google to develop its own solution, despite containerizing all services. In 2014, Google introduced Kubernetes, an open-source container orchestration tool evolved from its previous systems, Borg and Omega [4,5].

Kubernetes automates many tasks traditionally handled by system administrators, including infrastructure optimization, failover procedures, centralized logging, and monitoring. Additional functions and third-party applications developed through the Kubernetes API further enhance the essential features of Kubernetes, such as load balancing and auto-scaling. Kubernetes manages groups of one or more containers, called pods, handling deployment and resource control directly. However, Kubernetes’ default scheduler does not consider the coupling and dependencies of the containers, which can lead to inefficient resource usage and degraded performance in complex architectures [6]. To address these challenges, this paper proposes the Opportunistic Optimization Scheme for Pod Deployment (OOSP), a novel technique that optimizes pod placement by analyzing multi-layer data related to pod coupling and dependencies. By analyzing communication patterns between pods, the OOSP method reduces network latency by strategically placing frequently interacting pods on the same or nearby nodes, enhancing communication efficiency. Additionally, OOSP accounts for resource usage patterns by prioritizing resource-intensive pods, ensuring optimal allocation of CPU and memory across the cluster.

Experiments conducted in a virtualized Kubernetes environment using a microservice architecture benchmark application demonstrated significant performance improvements. Compared to the default Kubernetes scheduler, the proposed OOSP method reduced the average response time by 11.5% and increased the requests per second by 10.04%. These results indicate that the data-driven approach of OOSP minimizes communication delays between pods and improves overall system resource utilization. This research contributes a practical solution for optimizing performance and resource efficiency in cloud-native environments, addressing limitations in current Kubernetes scheduling strategies. We can generalize the OOSP approach to various cloud environments and workloads, providing a pathway for future research in container orchestration optimization.

This paper’s structure is as follows: Section 2 reviews the background and related work. Section 3 details the model design and implementation of the opportunistic pod placement optimization technique through system multi-layer inspection. Section 4 evaluates the proposed model’s performance by applying the algorithm to pod placement in a microservice architecture application. Section 5 examines potential malicious activities that could exploit the proposed optimization technique. Finally, Section 6 concludes the paper and discusses future research directions.

## 2. Related Work

This section reviews existing research on container scheduling, both general and Kubernetes-specific, highlighting their limitations and the distinctions of the system proposed in this paper.

### 2.1. Research on Container Scheduling

Wydrowski et al. [7] introduced Prequal, a novel load balancing approach for distributed multi-tenant systems. Unlike traditional methods focused on CPU load distribution, Prequal dynamically adapts to server heterogeneity and variable loads to minimize real-time request latency. It uses a probing mechanism to gather and process real-time server data, emphasizing Requests-in-flight (RIF) and response time. Deployed across various Google services, including YouTube, Prequal demonstrated significant improvements in latency, error rates, and resource utilization, proving more effective than traditional CPU-centric load balancing. However, Prequal’s asynchronous probing mechanism can lead to increased overhead with a higher number of probes, posing challenges in environments with limited CPU and network resources. Additionally, achieving optimal performance requires fine-tuning multiple parameters, which, if misconfigured, can degrade performance.

Safaryan et al. [8] proposed SLAM, a memory optimization method for serverless applications, focusing on Service Level Objectives (SLOs). SLAM aims to balance cost and performance by optimizing memory settings in a serverless computing environment. Unlike previous studies, which focused on cost optimization and SLO compliance for individual Function-as-a-Service (FaaS) functions, SLAM addresses multiple FaaS functions within complex application workflows. By employing distributed tracing, SLAM identifies relationships between FaaS functions and estimates execution times based on memory configurations to determine optimal settings. Experiments on AWS Lambda with four applications showed that SLAM’s configurations ensured over 95% of requests met predefined SLOs. However, increased the complexity in serverless applications can lead to tracing and modeling overheads, with large data volumes extending analysis time. Additionally, SLAM’s optimization algorithm may not fully account for interactions between memory configurations, potentially yielding lower performance improvements than expected in real applications.

Saidi et al. [9] discussed task scheduling and container deployment on physical nodes in cloud computing environments. Emphasizing the importance of optimizing task scheduling and container placement to enhance energy efficiency and overall system performance, the authors conducted a literature review from 2016 to 2023. They concluded that developing new scheduling algorithms capable of dynamically adapting to fluctuating cloud systems and efficiently handling requirements is essential.

Lavanya and Priya [10] proposed a multi-objective container scheduling and resource allocation approach using the Tuna Swarm Optimization (TSO) algorithm. The TSO algorithm, inspired by the swarm behavior of tuna fish in nature, is a metaheuristic algorithm specifically designed to optimize multiple objectives simultaneously. Container cloud environments, which require effective balance in resource allocation and task scheduling, particularly benefit from this approach. By using TSO, Lavanya and Priya were able to significantly reduce resource imbalances and improve response times. The algorithm simultaneously optimizes multiple objectives, including CPU and memory usage, and ensures the efficient deployment of containers to meet workload demands.

Zheng et al. [11] presented the GAIPPTSC method to address the orchestration and scheduling of containerized applications. The suggested solution employs a hybrid methodology that integrates a genetic algorithm (GA) with the Iterative Priority-based Preemptive Task Scheduling and Clustering (IPPTSC) algorithm. GAIPPTSC aims to minimize the execution duration of container tasks and decrease energy usage. Experimental findings indicate that GAIPPTSC realizes an average reduction in execution time of 24.7% and a decrease in energy usage of 53.6% relative to conventional container scheduling methods. Nonetheless, it may result in elevated communication expenses, and the algorithm’s execution duration can escalate by around fivefold relative to conventional methods.

Tang et al. [12] presented the Resource Overbooking and Container Scheduling (ROCS) algorithm to concurrently optimize resource overbooking and container scheduling in edge computing. ROCS employs a Soft Actor–Critic (SAC) reinforcement learning framework to enhance resource utilization while reducing the likelihood of errors related to container preemption and evictions. The proposed technique persistently evaluates the resource status of each edge node while simultaneously optimizing resource overbooking and scheduling decisions to improve the long-term profitability for cloud service providers. Experimental results demonstrate that ROCS improves resource utilization (CPU, memory, network bandwidth) by up to 75% and reduces the risk of overbooking compared to existing methods. Nonetheless, there exists a possibility of resource saturation resulting in container eviction, and the reinforcement-learning-based methodology may need substantial computational resources. Moreover, as the algorithm requires training time and data to adapt to new conditions, it may be unable to respond to sudden changes.

The algorithms in the previous studies mentioned above often focus on improving metrics such as memory and CPU utilization and focus on resource efficiency. The proposed OOSP technique is an algorithm that quantifies communication frequency and memory variation by jobs, then optimizing container placement based on the computed coupling and dependencies. OOSP aims to enhance resource efficiency and performance by assessing data across several layers and analyzing communication patterns and resource utilization among pods to facilitate optimal pod placement on nodes. In contrast to conventional scheduling algorithms that prioritize resource use, OOSP offers advanced placement tactics by taking into account the interactions across pods to determine optimal placement.

### 2.2. Research on Kubernetes Scheduling

Jung et al. [13] proposed a container orchestration placement technique to support the loose coupling of microservices. This study aims to improve performance by reflecting the characteristics of microservices when deployed in container orchestration frameworks. Utilizing frameworks like Kubernetes and Docker, the researchers presented a placement specification that considers the interconnectivity of microservices, enhancing performance, and reducing response latency. The study specifically designed the microservices architecture based on the Boundary-Control-Entity (BCE) pattern and incorporated this into Kubernetes deployment templates to maximize loose coupling characteristics. The results demonstrated the effectiveness of this approach in improving performance. However, the proposed technique’s reliance on extended templates to reflect specific microservice traits can increase complexity, requiring additional setup and management.

Goyal et al. (2021) [14] introduced a framework based on the Whale Optimization Algorithm (WOA) to optimize energy-resource allocation in cloud environments. This framework addresses the limitations of existing algorithms in balancing load, scheduling resources, and achieving energy efficiency. The researchers compared various optimization algorithms (PSO, CSO, BAT, CSA) and found that WOA significantly improved energy efficiency. In experiments, WOA outperformed other algorithms, demonstrating 0 ms response times and lower energy consumption (4536 J with seven servers and 8165.603 J with eight servers). Despite its superior performance in response time, energy consumption, execution time, and throughput, WOA’s computational complexity can increase in large-scale cloud environments, necessitating additional computational resources.

In Kubernetes, Ding et al. [15] proposed a method for microservice placement through dynamic resource allocation. This approach addresses the shortcomings of existing Kubernetes placement algorithms in managing dynamic resource contention and shared dependencies among microservice instances. The researchers created an integer nonlinear programming model for microservice placement. They wanted to keep costs as low as possible by figuring out how available each instance is and taking into account the total demand for resources when there is resource contention. They applied an improved genetic algorithm to this model. Experimental results showed higher throughput at the same cost. However, because genetic algorithms are so complicated, they might need more resources in large-scale cloud environments. And when there is a lot of dynamic contention, giving too many resources to the problem could slow it down.

Rossi et al. [16] proposed me-kube, a hierarchical scaling solution for Kubernetes-based microservice applications. Traditional horizontal scaling in Kubernetes uses system-oriented metrics like CPU utilization, which is inadequate for latency-sensitive applications. Me-kube introduces a hierarchical architecture where a central application manager coordinates subordinate microservice managers, allowing local control over microservice elasticity. To optimize performance, this approach employs pre- and post-hierarchical control policies based on queue theory. Experiments indicated that me-kube outperforms the default Kubernetes auto-scaler in response time and resource utilization. However, this hierarchical approach can create bottlenecks in central components, complicating management and scalability, especially in large systems. Additionally, designing and implementing the hierarchical architecture adds complexity.

Lai et al. [17] introduced a new scheduling algorithm known as Delay-Aware Container Scheduling (DACS), specifically designed to minimize latency in edge computing environments. DACS assesses applications’ latency sensitivity and calculates an acceptable delay range for each workload. The algorithm continuously monitors the real-time status of edge nodes, dynamically selecting the optimal node for container placement based on latency and node heterogeneity. By dynamically adjusting the scheduling decisions as network and node conditions change, DACS achieves significant reductions in latency when compared to the default Kubernetes scheduler. It is particularly effective for latency-sensitive applications, such as video streaming and real-time data processing, where even slight improvements in response time can lead to substantial performance gains.

In conclusion, while current research on container and Kubernetes scheduling addresses specific issues with distinct methodologies, challenges such as setup complexity, numerous parameters, overhead, and resource limitations persist in practical applications. Therefore, we need simpler, workload-specific container and Kubernetes scheduling techniques that we can directly apply to real workloads with minimal overhead.

As shown in Table 1, we can see that resource utilization and system performance can be improved simultaneously by considering the coupling and dependencies between pods, which have not been the main focus of previous studies. While existing methods focus on a single aspect, our proposed OOSP takes a multi-layered approach to achieve a more comprehensive optimization.

## 3. Model Design and Implementation

This section describes the design and implementation of the proposed opportunistic pod placement optimization system using multi-layer data sensing. It first examines the placement model and overall system architecture, followed by the workflow within each structure.

### 3.1. Overview of the Pod Placement Optimization Model

Designing a perfect container scheduling algorithm remains an unresolved issue, with many studies attempting various approaches to address its limitations. However, most research encounters practical application challenges and does not directly implement scheduling within Kubernetes [19].

Often, when designing and deploying applications using Kubernetes, the default scheduler and placement solutions fail to align with the characteristics of the running workloads or have low resource utilization for deployment [18]. Additionally, administrators manually specifying resource requirements may result in inaccurate settings, negatively impacting the entire cluster and leading to various human errors. To address these issues, administrators adjust the Kubernetes scheduler to suit operational service objectives, typically focusing on three main goals [20]:The optimization of workload performance.The optimization of resource utilization within the cluster.The reduction ion negative environmental impacts.

The proposed system in this paper focuses on optimizing workload performance among these three main goals. It implements scheduling that considers avoiding increased latency due to placing frequently communicating pods on distant nodes. The system identifies and assesses the coupling and dependencies of pods, recommending and applying pod placements to minimize human error resulting from inaccurate administrator settings.

The overall system architecture is shown in Figure 1. This system follows the basic architecture of Kubernetes, consisting of a master node with the Kubernetes Control-plane and worker nodes for pod deployment. Additionally, a sensing node is designed to collect multi-layer data from workloads without interfering with the performance of Worker nodes while still being managed within the same cluster using the Kubernetes API from the Master node.

### 3.2. Pod Placement Optimization Model Workflow

The workflow of the OOSP proposed in this paper is shown in Figure 2. First, the existing workload of the application is executed in the Run Workload module to identify what the service is doing in the state and which pods are used to perform the actions. Based on the information of the identified pods, the Data Collection module logs and extracts workload data at the cluster level and application level, which belongs to the Data Collection module, and preprocesses and parses them to the Data Analysis module to derive the coupling and dependency between pods. The Scheduling module suggests pod placement to the administrator based on the number of nodes to place the pods and the data derived, and the Orchestration module allows the administrator to perform orchestration by modifying the YAML file that deployed the existing workload in reference to the proposed pod placement [21]. YAML is a human-readable data serialization language used to write configuration files for infrastructure configuration and management.

### 3.3. System Architecture for Multi-Layer Data Sensing and Collection

The sensing node performs data collection at both the cluster level and application level within the Data Collection module to optimize pod placement. To ensure data are captured simultaneously, the cluster level and application level operate together. Each module collects data as follows.

#### 3.3.1. Cluster Level Data Sensing and Collection

The cluster level in the Data Collection module collects workload data, as shown in Figure 3. When scanning at the Cluster Level, queries are sent to Prometheus using the Kubernetes API to collect data. Prometheus detects and gathers data about workload services. Identified pods are monitored for individual metrics using Node Exporter Daemonsets for node state and performance metrics and cAdvisor functionality for container metrics. We aggregate all collected metrics back to Prometheus and then send them to the Sensing node via the Kubernetes API, where we store them as cluster.log data.

#### 3.3.2. Application Level Data Sensing and Collection

As shown in Figure 4, the application level in the Data Collection module collects workload data. When scanning at the application level, the Kubernetes API identifies the pods used by the service, and Tcpdump captures network traffic between pods. The Kubernetes API saves the captured traffic in pcap format and sends it to the Sensing node. We preprocess the saved pcap files to filter method calls invoked in the workload between nodes and store the resulting data as application.log.

### 3.4. System Architecture for Multi-Layer Data Analysis

Based on the collected data, the Data Analysis module performs analyses to optimize pod placement. The objective is to derive the coupling and dependencies between pods using application.log and cluster.log.

#### 3.4.1. Inferring Coupling Through Multi-Layer Data Analysis

Analysis of application.log, which contains preprocessed network traffic logs captured between pods, quantifies coupling between pods. Figure 5 shows the process flow for deriving coupling from application.log. The log data include timestamps, source IPs, destination IPs, HTTP response codes, HTTP methods, and brief packet contents. Kubernetes maps source and destination IPs to pod information to identify pod communication and count occurrences. Coupling is defined by the frequency of communication between pods during workload execution, with higher communication frequency indicating stronger coupling. We save the analyzed coupling data as a CSV file and visualize it to provide an intuitive understanding of workload coupling patterns.

#### 3.4.2. Inferring Dependencies Through Multi-Layer Data Analysis

We quantify dependencies between pods by analyzing cluster.log, which records memory usage logs for each pod during workload execution. Figure 6 shows the process flow for extracting dependencies from cluster.log. The log data include timestamps, pod names, nodes to which the pods belong, and memory usage. We rank pods based on their memory change rates by analyzing memory usage changes during workload execution. We prioritize pods with the highest memory change rates, assuming they can handle the most processing within the workload. We calculate dependency scores by summing the priority scores for each pair of pods, and then save the analyzed dependency data as a CSV file.

### 3.5. Pod Placement Optimization Algorithms

Based on the Data Analysis module’s results, this section presents the pod placement optimization algorithms using multi-layer data sensing proposed in this paper.

#### 3.5.1. Pod Placement Optimization Algorithm Using Coupling

Algorithm 1 describes the algorithm for optimizing pod placement based on the coupling between pods. First, step 1 determines the coupling degree. It is calculated based on the frequency of communication between pods. The coupling represents a value for how often pods communicate with each other. After the administrator enters the number of nodes to place, the function retrieves the coupling data from the CSV file and sorts them in descending order. Step 2 is to set a threshold value. Pods whose coupling value exceeds the calculated threshold (the average of all non-zero communication frequencies between pods) are considered strongly coupled and added to the list. Step 3 groups the pods. Pairs of pods that are measured to be strongly coupled are added to the list and sorted in order of coupling. We sequentially place the sorted pods from step 3 on nodes, rotating the excess pods between them. During this process, if we place a strongly coupled pod on a different node, we relocate it to the same node, prioritizing nodes with fewer pods. We calculate the coupling threshold by averaging all non-zero values in the data structure. This algorithm produces an optimal placement of pods based on the number of nodes specified by the administrator.
**Algorithm 1** Suggest optimal pod placement using coupling**Require:** 
DataFrame *df_counts*, List of Nodes *nodes*, Mean Threshold *mean_threshold***Ensure:** 
Optimal Pod Placement *placement*1: sorted_pods← Sort pods by total communication count in descending order2: **for** each pod1 in df_counts.index **do**3:    **for** each pod2 in df_counts.columns **do**4:       **if** pod1≠pod2 **and** df_counts[pod1,pod2]>mean_threshold **then**5:          Append (pod1,pod2,df_counts[pod1,pod2]) to pod_coupling6:       **end if**7:    **end for**8: **end for**9: Sort pod_coupling by communication count in descending order10:**for** each i,pod in sorted_pods_coupling **do**11:   node←nodes[i%len(nodes)]12:   Append pod to placement[node]13:**end for**14:**for** each (pod1,pod2,count) in pod_coupling **do**15:   node1← Node containing pod1 in placement16:   node2← Node containing pod2 in placement17:   **if** node1≠node2 **then**18:     **if** len(placement[node1])<len(placement[node2]) **then**19:        Move pod2 to node120:        Remove pod2 from node221:     **else**22:        Move pod1 to node223:        Remove pod1 from node124:     **end if**25:   **end if**26:**end for**27:**return** 
placement

#### 3.5.2. Pod Placement Optimization Algorithm Using Coupling and Dependencies

The algorithm for optimizing pod placement using both coupling and dependencies inferred from memory usage changes is shown in Algorithm 2. The administrator inputs the number of nodes, and the function references the coupling and dependency CSV files, normalizing the values between 0 and 1. Weighted values are applied to these normalized data to create a new data structure, with the sum of weights equaling 1. The weights can be adjusted by the administrator based on the importance of communication efficiency, assigning a higher weight to coupling, or memory efficiency, assigning a higher weight to dependencies. Pods are sorted by their coupling values, with strongly coupled pod pairs added to dependency groups, which are also sorted. Pods are then placed on nodes in sequence, with excess pods being circulated among nodes. If dependent pod pairs are placed on different nodes, they are relocated to nodes with fewer pods. This algorithm returns an optimized pod placement that considers both coupling and dependencies, reducing communication latency and enhancing overall system performance.
**Algorithm 2** Suggest optimal pod placement using coupling and dependency**Require:** 
DataFrame counts_df, DataFrame dep_df, List of Nodes nodes, Mean Threshold mean_threshold, Weight *X*, Weight *Y***Ensure:** 
Optimal Pod Placement placement1: norm_counts_df←counts_df/counts_df.values.max()2: norm_dep_df←dep_df/dep_df.values.max()3: combined_df←X×norm_counts_df+Y×norm_dep_df4: sorted_pods← Sort pods by total communication and coupling count in descending order5: **for** each pod1 in combined_df.index **do**6:     **for** each pod2 in combined_df.columns **do**7:       **if** pod1≠pod2 **and** combined_df.at[pod1,pod2]>mean_threshold **then**8:          Append (pod1,pod2,combined_df.at[pod1,pod2]) to pod_dependencies9:       **end if**10:   **end for**11:**end for**12:Sort pod_coupling by combined count in descending order13:**for** each i,pod in enumerate(sorted_pods) **do**14:   node←nodes[i%len(nodes)]15:   Append pod to placement[node]16:**end for**17:**for** each (pod1,pod2,count) in pod_coupling **do**18:   node1← Node containing pod1 in placement19:   node2← Node containing pod2 in placement20:   **if** node1≠node2 **then**21:     **if** len(placement[node1])<len(placement[node2]) **then**22:        Move pod2 to node123:        Remove pod2 from node224:     **else**25:        Move pod1 to node226:        Remove pod1 from node127:     **end if**28:   **end if**29:**end for**30:**return** 
placement

#### 3.5.3. Threshold Calculation Algorithm for Pod Placement Optimization

In Algorithm 3, the algorithm for calculating the threshold used in the pod placement optimization algorithms is shown. Algorithm 3 computes the threshold by averaging all non-zero values in the data structure. The algorithm flattens the data structure into an array, filters out zero values, and calculates the average of the remaining values. The pod placement optimization algorithms then evaluate coupling and dependencies using this calculated average threshold. Using an average threshold provides a balanced and adaptive mechanism for determining which pods should be prioritized for optimized placement in a dynamic Kubernetes environment, with a balanced degree of coupling and reliance on the data collected.
**Algorithm 3** Calculate mean threshold**Require:** 
DataFrame df_counts**Ensure:** 
Mean Threshold mean_threshold1:all_counts←df_counts.values.flatten()2:non_zero_counts←all_counts[all_counts>0]3:$mean\_threshold\leftarrow \frac{1}{n}\sum_{i=1}^{n}x_{i}\;\mathrm{for}\;x_{i}\in non\_zero\_counts$4:**return** 
mean_threshold

## 4. Experiments and Performance Evaluation

This section evaluates the performance of the proposed model. We implemented the model on nodes deployed with Kubernetes in a virtualized environment, managing each node as a single cluster. We used Linux as the operating system and conducted the performance evaluation using a web application with a microservice architecture.

### 4.1. Evaluation Metrics

**Average response time**: The proposed model aims to improve performance by optimizing the pod placement of applications deployed on Kubernetes. Average response time measures the time taken for the application to complete a user’s request. This metric helps determine if the new pod placement reduces latency and enhances the user experience. The speed of processing users’ requests is a crucial performance metric in a cloud-native environment. The average response time directly reflects the user experience and measures how fast the system processes requests with the given resources. This was directly related to the goal of the proposed technique to reduce response time by reducing inter-pod communication latency and optimizing resource utilization.

**Requests per second (Requests/s):** This metric indicates the number of requests the system or server can handle per second. The experiment’s web application processes requests like loading web pages and making API calls. This metric demonstrates whether the proposed model’s pod placement changes improve processing capabilities and handle high traffic volumes more effectively. We believe that the number of requests a system can process in a given amount of time is an important metric for evaluating a system’s processing power. By increasing the number of requests per second, we can see how the proposed technique contributes to maximizing cluster resource utilization and increasing throughput.

### 4.2. Experimental Setup

Table 2 describes the experimental setup for evaluating the performance of the proposed model. We deployed Docker on each node to leverage the container system. The Kubernetes control plane was hosted on the Master node, while Kubernetes clients were installed on the Worker and Sensing nodes, forming a managed cluster.

#### 4.2.1. Benchmark Applications

To evaluate the model, we selected *Teastore* [22], an e-commerce platform developed to study microservice operations. *Teastore* is a representative benchmark application based on a microservice architecture. Figure 7 illustrates the architecture of *Teastore*.

Our research goals focus on optimizing performance and improving resource efficiency. Response time and throughput are the most appropriate metrics to evaluate our goals. These metrics provide a snapshot of the overall resource utilization and performance of the system.

*Robot-shop* [23] was also selected for evaluation. It is an online application for selling robots, consisting of 12 microservices. The architecture of *Robot-shop* is shown in Figure 8.

To simulate actual workloads of microservices, we used the *Locust* [24] load testing tool. *Locust* allows administrators to write and execute load tests, simulating user behavior. The *Locust* script for the experiment involves users visiting the shop page, logging in with a random username, browsing, adding items to the cart, making purchases, and visiting the profile page to log out. We repeat this sequence to generate the load.

To simulate actual workloads of microservices, we used the *ApacheBench* [25] benchmarking tool. ApacheBench is a widely used web service benchmarking tool that measures the performance of HTTP servers. The tool can generate significant load by sending numerous requests to a web server, thereby simulating high traffic. For our experiment, ApacheBench was configured to send a series of HTTP requests to the *Teastore* and *Robot-shop* application, measuring metrics such as the number of requests per second, the time per request, and the transfer rate. This setup allowed us to assess the system’s performance under varying levels of load and identify potential bottlenecks.

#### 4.2.2. Experimental Configuration

The experimental configuration for evaluating the proposed model’s performance involves deploying application pods on Worker nodes from the Master node using Kubernetes’ default scheduler. Figure 9 illustrates the deployment of the *Teastore* application on Worker nodes using the default Kubernetes scheduler. We excluded policies like NodeSelector or NodeAffinity and deployed pods using YAML files with default configurations. To gather actual workload data, the *Locust* load testing tool was used to simulate five virtual users operating for one minute. Simultaneously, the Data Collection module performed scans at both the cluster level and the application level.

The administrator then calculated the coupling and dependencies to determine the pod placement. The administrator manually applied the placement recommendations of the proposed model to the YAML files for orchestration. In the experiment, the weights for Algorithm 2 were set to 0.3 for coupling (X) and 0.7 for dependency (Y), assuming a workload prioritizing memory efficiency.

### 4.3. Experimental Results and Performance Evaluation

This section compares the performance of the proposed pod placement algorithms using the *Teastore* and *Robot-shop* applications. The scenarios include Native, which uses only Worker nodes without Kubernetes; Kubernetes, which uses Kubernetes’ default scheduler; Count Coupling, which applies Algorithm 1 from this study; and Resource Opportunistic, which applies Algorithm 2 from this study. To evaluate the proposed model, we measured the average response time and requests per second for each pod placement scenario using a load tester simulating the behavior of five virtual users over two minutes.

#### 4.3.1. Pod Placement Changes by Algorithm

Figure 10 shows the pod placement results for the benchmark application *Teastore* with the default scheduler and the two proposed algorithms. It shows the initial pod placement of the Teastore application using the default Kubernetes scheduler called Kubernetes State and the modified pod placement by Algorithm 1, which improves communication efficiency by placing pods based on coupling, and Algorithm 2, which optimizes resource efficiency and communication efficiency simultaneously by considering coupling and dependency together.

Algorithm 1 measures coupling based on the frequency of communications and reflects this in the pod placement. Algorithm 2 prioritizes pods based on the memory changes observed in the workload, calculating dependencies and incorporating these into the pod placement.

Figure 11 shows the change in pod placement for the benchmark application *Robot-shop*, starting with the default Kubernetes pod placement and applying Algorithm 1, which improves communication efficiency, and Algorithm 2, which optimizes communication efficiency and resource efficiency.

#### 4.3.2. Performance Evaluation

To evaluate the performance of the proposed model, we used a load tester that simulates the behavior of actual workloads. We measured the average response time and requests per second for each pod placement scenario. The load tester was configured to simulate five virtual users operating the application for two minutes.

Figure 12 shows the change in average response time for each pod placement method in the *Teastore* application. We compared the default Kubernetes scheduler with the proposed algorithms (Algorithms 1 and 2). Because the default scheduler does not consider the communication patterns between pods, pods with high communication frequencies are placed on different nodes, causing network delay. As a result, the average response time increases. Algorithm 1 places pods according to their degree of coupling, reducing network delay by placing pods with high communication frequencies on the same node. As a result, we see an 8.61% reduction in average response time. Algorithm 2 considers dependencies (memory usage) in addition to coupling to place resource-intensive pods. This optimizes not only communication efficiency but also resource usage efficiency, improving the average response time by 1.49%.

Figure 13 shows the change in requests per second for each method of pod placement in the *Teastore* application. It shows how many requests the system can handle. The default scheduler fails to optimize resource usage and communication patterns, resulting in network latency, which limits the number of requests per second it can handle. By applying Algorithm 1, highly coupled pods are placed on the same node, which reduces network latency and allows more requests to be served. This results in an 8.7% increase in requests per second. Algorithm 2 takes resource usage patterns into account, optimizing both communication and resource efficiency, resulting in a 1.49% increase in requests per second. These results indicate that count coupling is a more efficient orchestration method for the *Teastore* workload, emphasizing low resource usage and communication efficiency [26].

Figure 14 shows a graph comparing the average response time for each pod placement scheme in the *Robot-shop* application. Because the default scheduler does not consider communication patterns and resource efficiency, the average response time is high due to communication delays between pods. By applying Algorithm 1, pods with high communication frequency are placed on the same node, which reduces network delay, resulting in a 7.61% improvement in average response time. Algorithm 2 optimizes the balance of resources and communication by considering both resource usage and coupling, resulting in an 11.5% improvement in average response time.

Figure 15 shows the change in requests per second for each arrangement for the *Robot-shop* application. The default Kubernetes scheduler’s placement does not take into account resource usage efficiency and communication patterns, resulting in a relatively low number of requests per second. Algorithm 1 increases the number of requests processed per second by 7.49% by optimizing communication between pods to reduce network latency. Algorithm 2 optimizes not only communication but also resource usage, showing a 10.04% increase in requests per second. These results indicate that Resource Opportunistic is a more efficient orchestration method for the *Robot-shop* workload, emphasizing resource efficiency over communication efficiency [27]. From our experiments with the two applications above, we can see that pod placement using Algorithms 1 and 2 performs better than traditional Kubernetes scheduler placement, and we can use resources more efficiently to obtain better performance with the same amount of resources.

#### 4.3.3. Limitations of the Experiment

However, since the experiments were conducted in a specific cloud environment and benchmark application setup, these settings may not reflect the many variables and complex situations that may occur in real-world diverse cloud infrastructures. Therefore, further research is needed to generalize to a variety of cloud environments and applications. For example, different network architectures, resource allocation schemes, and hardware configurations used by different cloud providers may lead to performance differences, and different workloads or application types may exhibit different responses, so further experiments should validate the proposed techniques on different infrastructures such as public, private, and hybrid environments.

In addition, you can think about what happens when the number of nodes and pods increases, which is one of the hallmarks of cloud systems: flexible scaling up and down. In the proposed technique, when the number of pods increases (scale up), when a pod with the same purpose is created within a node, it is possible to track and collect data from the time it is created. This is not the behavior of a normal application, but it shows that it is designed to track the behavior of pods during scale-up.

If a pod failed during application operation, we configured the *ReplicaSet* to track copies of the pod created through the same image that is recovered through the *ReplicaSet* so that we could collect data from the previously failed pod and the newly recovered pod to obtain qualitative results. Again, this is not the behavior of a normal application, but it shows that we designed it so that we can track the behavior of a Pod in the event of a failure.

#### 4.3.4. Discussion on the Scalability of the Algorithm

The algorithm of the OOSP proposed in this paper was implemented on a small to medium-sized Kubernetes cluster. However, we also need to consider the possibility of scaling to larger Kubernetes clusters.

**Scalability:** The algorithms within OOSP are designed to optimize placement by analyzing the coupling and dependencies between pods based on multi-tiered data, allowing data analysis and placement decisions to be done in parallel as the cluster size grows, provided the overhead of data collection is addressed. In addition, the Kubernetes API allows you to monitor the health of each node and pod in real time, so you can maintain efficient resource management even in large clusters. However, in this paper, the OOSP algorithm is tailored for small and medium-sized Kubernetes clusters, and the part about large-scale scalability is a consideration for future research. The current algorithm is difficult to apply to complex distributed systems, so it must be modified to apply to separate large-scale systems.

**Calculation complexity:** In the proposed algorithm, the computation required to compute the coupling degree increases as the cluster size increases, but this can be optimized by applying distributed processing techniques and clustering algorithms to the algorithm proposed in this paper.

**Realizability:** As the number of nodes and pods increases, there is a possibility of network latency or communication overhead. Designed to maintain communication efficiency and resource utilization in large environments, the proposed technique addresses the overhead of data collection in large clusters by placing highly coupled pods on physically close nodes and applying a deployment strategy that takes resource utilization patterns into account.

**Real-time pod scaling:** OOSP algorithms aim to capture adaptive data and schedule pods based on workload characteristics rather than focusing on real-time fluctuations. Nonetheless, the techniques utilized in “real-time data streaming” can be applied to achieve real-time [28,29]. The aim of real-time data streaming is to decrease latency and ensure a continuous data flow by reducing network latency and optimizing bandwidth efficiency. Similarly, the proposed OOSP reduces the frequency of matrix collection and transfers the gathered data to a separate analytics server to enhance scheduling by utilizing computed values to swiftly adjust pod placement while minimizing reaction time. The OOSP can achieve real-time objectives by using the concept of continuous data stream processing, which involves dynamically adjusting data collection intervals based on workload intensity and utilizing parallel processing techniques for matrix collection and computation on the analytics server. Nevertheless, achieving this realism results in an increase in system burden and complexity, as it necessitates the allocation of additional processing power and resources to manage real-time computation and decision-making, as well as to handle more frequent data cycles. Further research is necessary to address these issues, which are comparable to the obstacles encountered by real-time network management systems. This research should focus on reducing system burden, enhancing processing speed, and increasing resource efficiency.

## 5. Security Implications and Vulnerabilities in Pod Deployment

### 5.1. Possible Administrative and Technical Vulnerabilities in Pod Deployments

**Increased attack surface:** If pods are placed together on the same node with a high degree of coupling, a malicious user can use an attack on one pod to affect neighboring pods. In particular, side-channel attacks can be used to exploit vulnerabilities between pods that frequently communicate with the network.

**Vulnerabilities on shared resources:** When multiple pods share the same resources (CPU, memory, etc.) during resource optimization, certain pods may overuse resources and cause DoS internally, which can be exploited by malicious users to crash the application.

**Network vulnerabilities:** Placement is based on the frequency of communication between pods, which can lead to an increase in network intrusion attack attempts on nodes with concentrated communication paths.

### 5.2. Administrative and Technical Vulnerabilities with Optimization Techniques

While optimization algorithms are focused on maximizing performance, they can introduce vulnerabilities based on resource placement and communication patterns. For example, when optimizing the placement of pods based on resource usage and communication frequency, a focus on optimization alone can concentrate the attack surface on certain nodes.

### 5.3. Security Hardening Measures

**Pod isolation and network segmentation:** Even if highly coupled pods are placed together, each pod can be isolated at the network level or network segmentation using subnets can be used to propose direct communication between pods. This reduces the risk of side-channel attacks or network intrusions.

**Monitoring resource allocation:** Processes or security mechanisms should be in place to monitor resource usage to detect resource exhaustion attacks or unusual resource usage patterns.

**Deploy security awareness:** security-aware placement techniques should be added using security-embedded algorithms to optimize applications for performance while also introducing security enhancements.

## 6. Conclusions

In this study, we presented an opportunistic pod placement optimization technique using multi-layer data sensing to improve resource utilization and system performance in Kubernetes environments. The method, which considers both pod communication efficiency and resource usage, significantly reduced average response time by up to 11.5% and increased requests per second by up to 10.04%, outperforming the default Kubernetes scheduler.

The key contribution of this work is the integration of communication patterns and resource dependencies into the pod placement process, which enhances both performance and efficiency. While the experiments were conducted in controlled environments, future research should focus on validating the approach in more dynamic cloud settings and exploring real-time optimization techniques to further enhance system responsiveness.

Future work will focus on applying the proposed methodology to real-time cloud systems with dynamically changing workloads and resource demands, and will augment the technique with environmental variables such as workload, complex workloads, network, and resource contention found in real cloud environments. Furthermore, investigating the scalability of the approach in large-scale cloud environments and its ability to handle live pod migration without service interruption is essential for further validation.

## Figures and Tables

**Figure 1 sensors-24-06244-f001:**
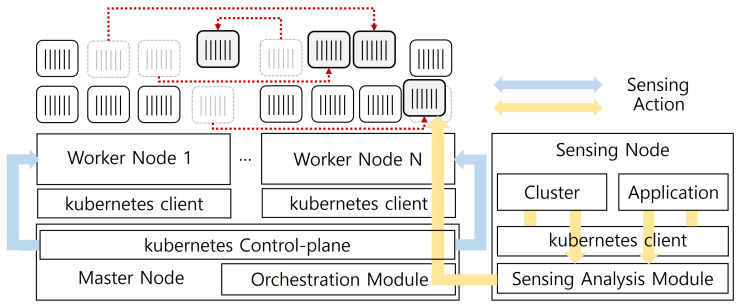
Overall system architecture.

**Figure 2 sensors-24-06244-f002:**
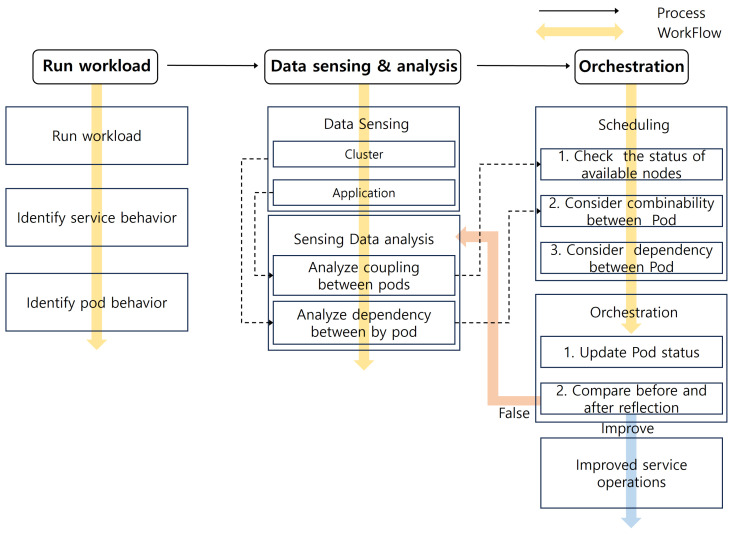
Overall System process flow.

**Figure 3 sensors-24-06244-f003:**
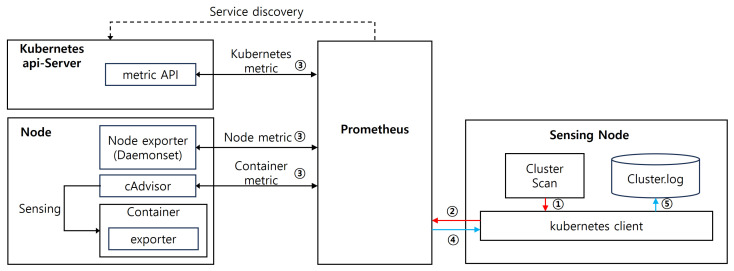
Cluster Level data collection structure.

**Figure 4 sensors-24-06244-f004:**
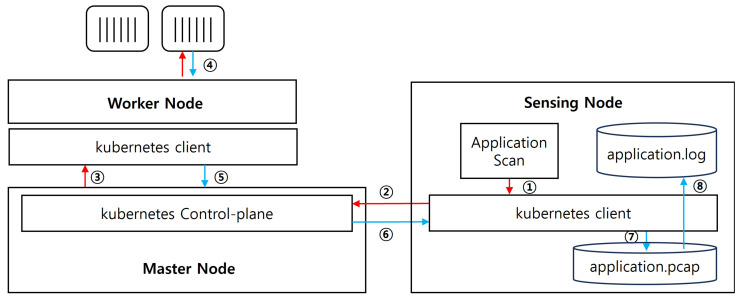
Application Level data collection structure.

**Figure 5 sensors-24-06244-f005:**
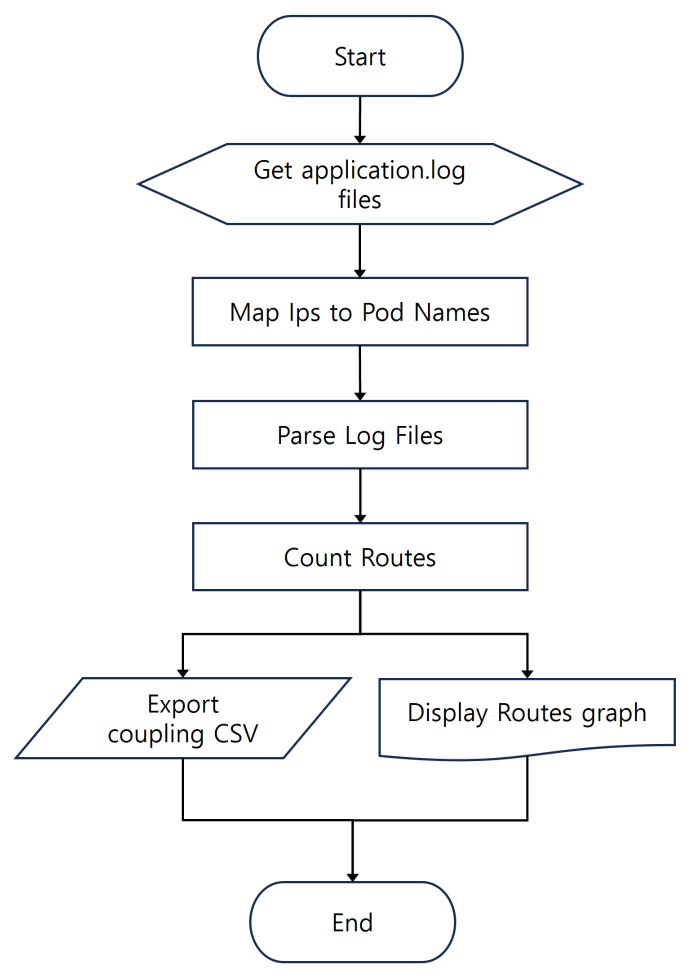
Cohesion Inference Process Flowchart.

**Figure 6 sensors-24-06244-f006:**
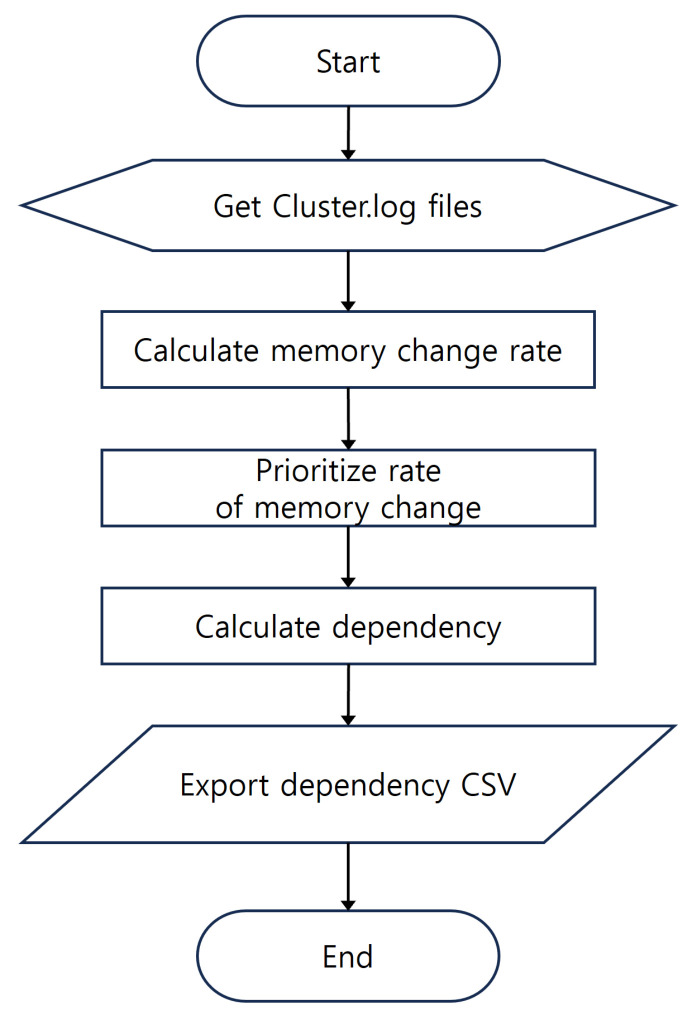
Dependency inference process flowchart.

**Figure 7 sensors-24-06244-f007:**
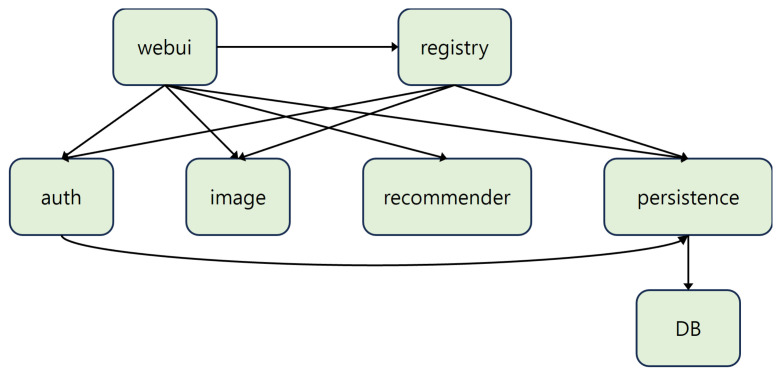
Teastore architecture.

**Figure 8 sensors-24-06244-f008:**
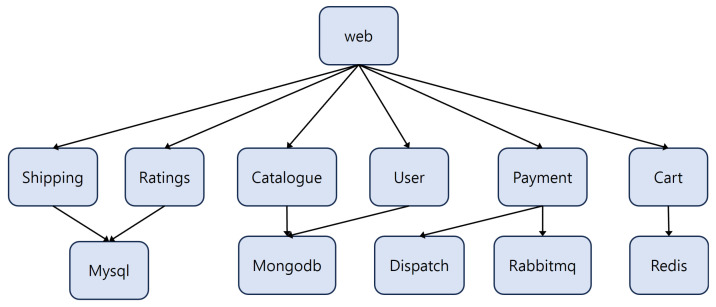
Robot-shop.

**Figure 9 sensors-24-06244-f009:**
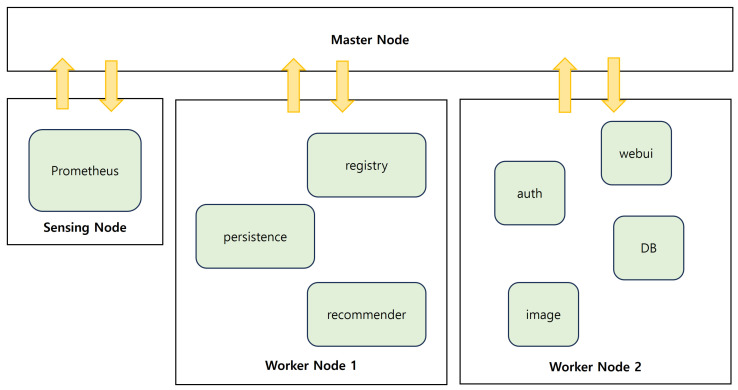
Example of pod placement using a Kubernetes scheduler.

**Figure 10 sensors-24-06244-f010:**
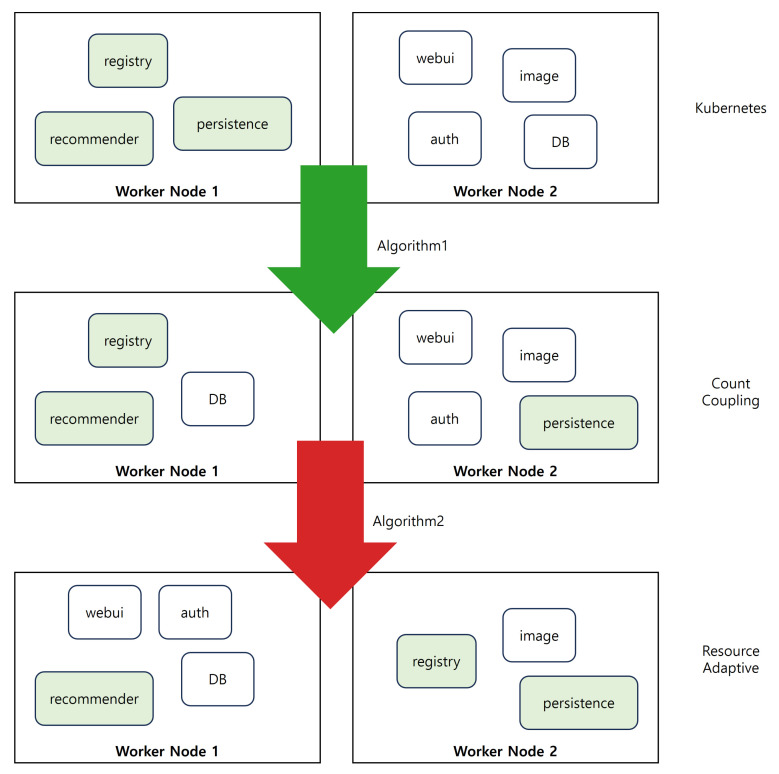
Changes in *Teastore* pod placement by the algorithm.

**Figure 11 sensors-24-06244-f011:**
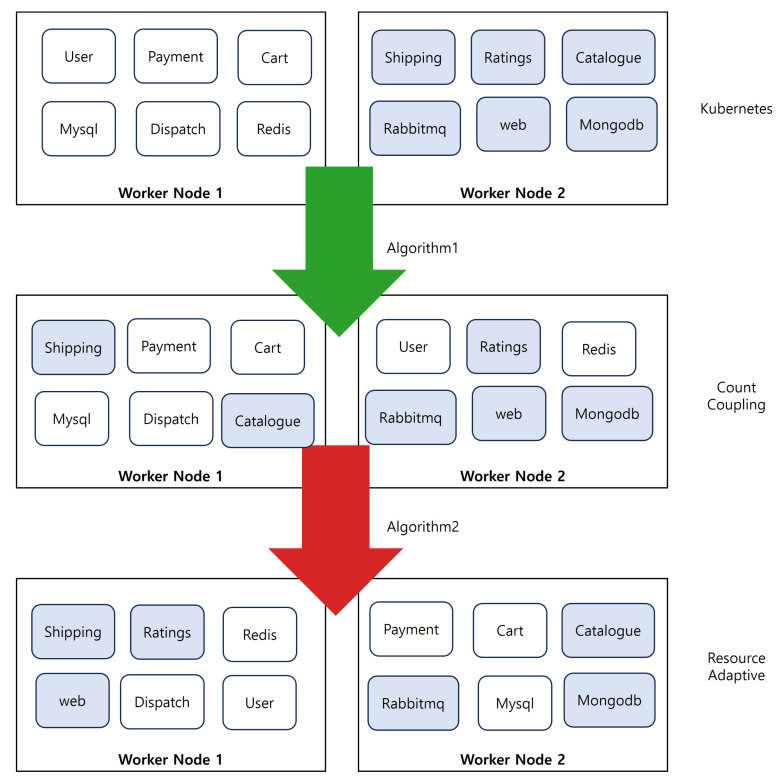
Changes in *Robot-shop* pod placement by the algorithm.

**Figure 12 sensors-24-06244-f012:**
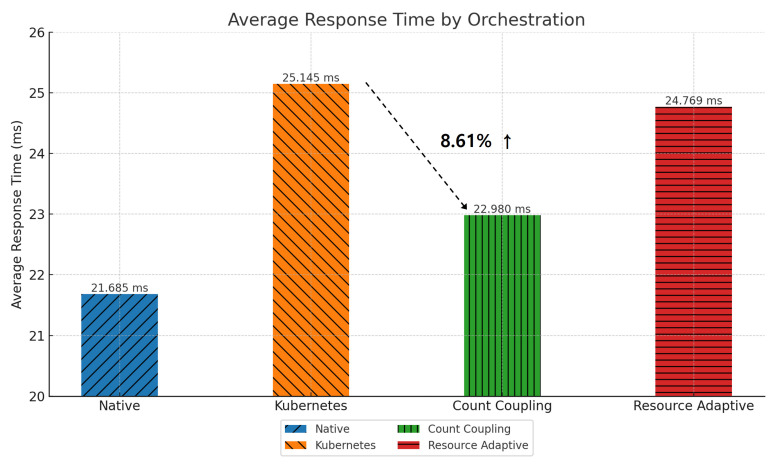
Average response time by pod placement in *Teastore*.

**Figure 13 sensors-24-06244-f013:**
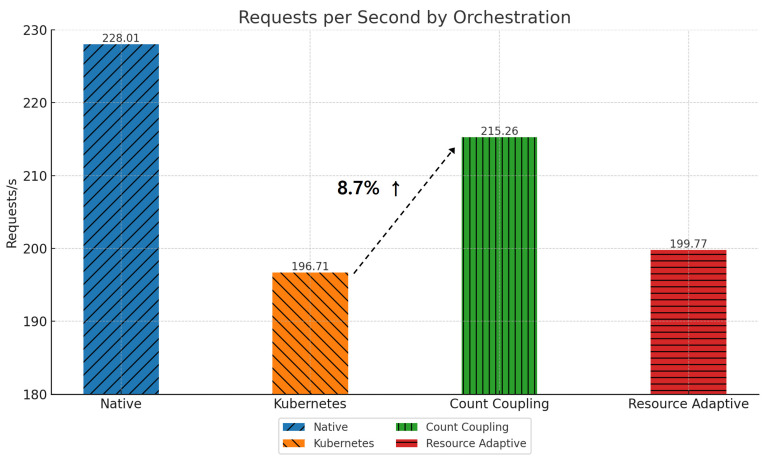
Requests per second by pod placement in *Teastore*.

**Figure 14 sensors-24-06244-f014:**
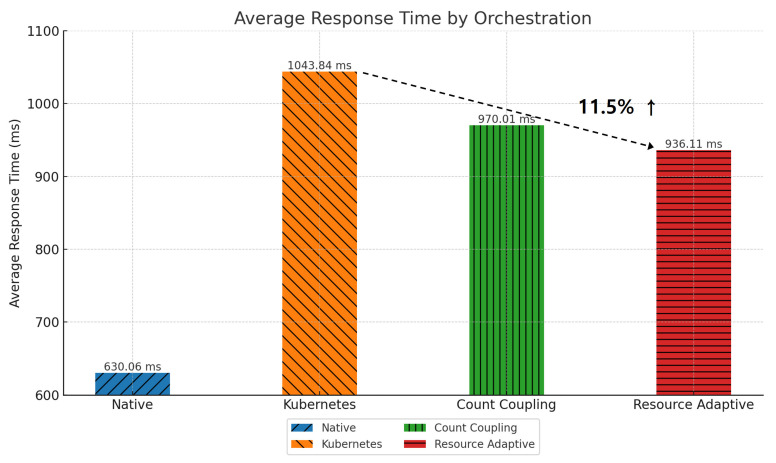
Average response time by pod placement in *Robot-shop*.

**Figure 15 sensors-24-06244-f015:**
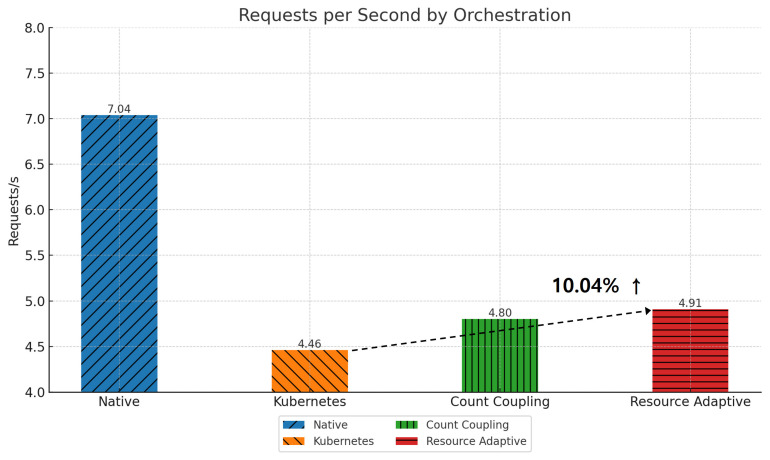
Requests per second by pod placement in *Robot-shop*.

**Table 1 sensors-24-06244-t001:** Analysis of existing Service Scheduling research results.

Research	Purpose	Algorithms	Performance
[8]	Memory Optimisation	SLO-aware Techniques	Optimise memory usage
[9]	Job Scheduling and VM Placement	-	Speed up task processing
[10]	Container cloud resource allocation and scheduling	Tuna Swarm Optimization (TSO)	Reduce resource imbalance and improve response time
[11]	Container Tasks Scheduling	GA (Genetic Algorithm) + IPPTSC	Reduced execution time and energy
[12]	Resource Overbooking and Container Scheduling	SAC Reinforcement Learning	Increased resource utilization, reduced eviction risk
[13]	Deploying Microservices	-	Improve scalability and maintainability
[14]	Energy-Resource Allocation	Whale Optimisation Algorithm	Increase energy efficiency
[15]	Deploying Microservices	Dynamic Resource Allocation Algorithms	Increased resource utilisation
[16]	Scaling Microservices	Hierarchical Scaling	Efficient scaling
[17]	Optimize container placement based ondelay sensitivity	DACS (Delay-Aware Container Scheduling)	Reduce latency and improve real-time application performance
[18]	Scheduling deep learning jobs	Intelligent Resource Estimation	Improved performance and increased network efficiency
OOSP	Optimising Pod Placement	Multilayer Data Analytics	Reduced response times and increased throughput

**Table 2 sensors-24-06244-t002:** Experimental system environment.

Master Node	
CPU Cores	8
Memory	16 GB
Operating System	Ubuntu 20.04.6 LTS
Docker version	26.0.0
Kubernetes version	1.28.8
**Worker, Sensing Node**	
CPU Cores	4
Memory	4 GB
Operating System	Ubuntu 20.04.6 LTS
Docker version	26.0.0
Kubernetes version	1.28.8

## Data Availability

The original contributions presented in the study are included in the article, further inquiries can be directed to the corresponding author.

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
