# Peer review of "OOSP: Opportunistic Optimization Scheme for Pod Deployment Enhanced with Multilayered Sensing"

_sensors, 2024, doi:10.3390/s24196244_

Round 1

Reviewer 1 Report

Comments and Suggestions for Authors

The authors proposed an opportunistic pod placement optimization technique using multi-layer inspection to address the issues such as performance degradation and resource wastage in default scheduler, optimized for CPU and memory usage.  The method analyzes multi-layer data to derive optimal placements based on pod coupling and dependencies, thereby improving resource utilization and performance. However, the manuscript has adopted a novel approach to prove its place, still some basic observations need proper justifications. I suggest the authors to work on the following issues:

1.      The content of the abstract is structured well, and the importance of the research problem is clearly stated. However, the abstract has to be revised and concise. It is better to mention the all-validation metrics used to compare the performance of the proposed technique with that of state-of-the-art techniques

2.      Introduction should be shortened by removing redundant descriptions and the major contributions of the proposed model should be discussed in detail (i.e.) describe the efficiency of opportunistic pod placement optimization technique in stepwise pattern with the corresponding data and the performance upgradation of a proposed model comparing with the existing model

3.      In the related work section, the authors can include recent works on container scheduling and Kubernetes scheduling.

·         Lai, W.K., Wang, Y.C. and Wei, S.C., 2023. Delay-aware container scheduling in kubernetes. IEEE Internet of Things Journal10(13), pp.11813-11824.

·         Lavanya, J. and Kavi Priya, S., Multi-objective comprehensive container scheduling and resource allocation for container cloud using tuna swarm optimization algorithm. Journal of Intelligent & Fuzzy Systems, (Preprint), pp.1-17.

·         Lai, W.K., Wang, Y.C. and Wei, S.C., 2023. Delay-aware container scheduling in kubernetes. IEEE Internet of Things Journal10(13), pp.11813-11824.

4.      In Section 3.2, the authors haven’t discussed the Run Workload module of the overall system process flow depicted in Figure 2. A detailed description of the Run Workload module should be included, along with its corresponding identification of service behavior and pod behavior. This addition will provide a more comprehensive understanding of the system's operation and dynamics.

5.      The authors have mentioned "YAML files" in the context of modifying the workload and executing the orchestration in Section 3.2. However, a detailed explanation of YAML files is necessary for a better understanding, especially for researchers in the cloud computing environment. Also, citing relevant manuscripts or references regarding YAML would provide further clarity and support.

6.      The line 253 clarity is missing

7.      The authors are requested to illustrate Algorithm 1 and Algorithm 2 in detail. Providing comprehensive explanations and stepwise descriptions of each algorithm will enhance the clarity for the researchers

8.      The authors can include the mean threshold formula in threshold calculation for Pod Placement Optimization

9.      Why the authors have considered only two performance metrics (average response time and request per second) to ensure the performance of the proposed model.

10.  Is it possible to evaluate the resource utilization before and after deploying the optimization algorithm to ensure that the resources are efficiently used?

11.   The authors can include latency and throughput to ensure the application-level performance of the proposed model

12.  It's better to elaborate on the inferences of Figure 10 and Figure 11

13.  The inferences of figure 12,13,14,15 to be enhanced by explaining the advantage of improved average response time and request per second

14.  Improve the clarity of all figures

15.  Check the typos and grammatical errors in the entire manuscript

16.  The language of the paper should be revised, there are multiple errors and the text is often unclear

17.  Conclusion section must be revised and precise

                I would recommend a “Major revision” to enhance the quality of the article and would like to review an updated version of the manuscript based on the above-said comments. 

Comments on the Quality of English Language

Can be improved.

Reviewer 2 Report

Comments and Suggestions for Authors

Comments on the Quality of English Language

"The proposed method reduced average response time by up to 11.5% and increased requests per second by up to 10.04% compared to the default Kubernetes scheduler." Line 9

"The dynamic nature of cloud-native systems requires continuous adjustments by system architects, increasing administrative fatigue." Line 50

"Additionally, administrators manually specifying resource requirements may result in inaccurate settings, negatively impacting the entire cluster and leading to various human errors." Line 171

 "The Application Level in the Data Collection module collects workload data as shown in Figure 4." Line 214

"The saved pcap files are preprocessed to filter method calls invoked in the workload between nodes." Line 219

"In modern cloud environments, container orchestration tools are essential for managing workloads and services." Add "the".

"Additionally, administrators manually specifying resource requirements may result in inaccurate settings, negatively impacting the entire cluster and leading to various human errors."

Round 2

Reviewer 2 Report

Comments and Suggestions for Authors

The authors have taken into consideration all remarks. Ultimately, the manuscript has been significantly improved. 

However, the section of background & related work should be imroved. Include more Container Scheduling Techniques:

Some State-of-the-art methods are:

i) Prequal (Load Balancing for Distributed Systems).

ii) SLAM (Memory Optimization for Serverless Applications).

iii) Tuna Swarm Optimization (TSO).

(+) It would be usefull if you compare the proposed frame (OOSP) with other existing methods in terms of (e.g., memory, CPU usage).

-) Real-time Pod Scaling: Discuss how OOSP could scale in real-time. Explain this better.

Comments on the Quality of English Language

Minor.

Author Response

Reply Letter

  • Manuscript ID: sensors-3132164
  • Manuscript Title: “OOSP: Opportunistic Optimization Scheme for Pod Deployment Enhanced with Multilayered Sensing”
  • Authors: Joo-Young Roh, Sang-Hoon Choi, and Ki-Woong Park

September 21, 2024

Dear Reviewers of this manuscript:

We wish to submit our revised manuscript reflecting the reviewers’ comments regarding publication in SENSORS, titled “OOSP: Opportunistic Optimization Scheme for Pod Deployment Enhanced with Multilayered Sensing”

We would like to thank the reviewers for their time and valuable comments. We thoroughly revised the manuscript in accordance with your comments, and our responses are enclosed. All of your comments were helpful toward improving the quality of this paper.

Thank you.

Sincerely,

Prof. Dr. Ki-Woong Park (Corresponding Author)
